# Fabrication of Multi-Material Pneumatic Actuators and Microactuators Using Stereolithography

**DOI:** 10.3390/mi14020244

**Published:** 2023-01-18

**Authors:** Qingchuan Song, Yunong Chen, Peilong Hou, Pang Zhu, Dorothea Helmer, Frederik Kotz-Helmer, Bastian E. Rapp

**Affiliations:** 1Laboratory of Process Technology, NeptunLab, Department of Microsystems Engineering (IMTEK), University of Freiburg, Georges-Köhler-Allee 103, 79110 Freiburg, Germany; 2FIT Freiburg Center of Interactive Materials and Bioinspired Technologies, University of Freiburg, 79110 Freiburg, Germany; 3Glassomer GmbH, In den Kirchenmatten 54, 79110 Freiburg, Germany; 4Freiburg Materials Research Center (FMF), University of Freiburg, 79104 Freiburg, Germany

**Keywords:** multi-material 3D printing, stereolithography, pneumatic actuator, microactuator

## Abstract

Pneumatic actuators are of great interest for device miniaturization, microactuators, soft robots, biomedical engineering, and complex control systems. Recently, multi-material actuators have become of high interest to researchers due to their comprehensive range of suitable applications. Three-dimensional (3D) printing of multi-material pneumatic actuators would be the ideal way to fabricate customized actuators, but so far, this is mostly limited to deposition-based methodologies, such as fused deposition modeling (FDM) or Polyjetting. Vat-based stereolithography is one of the most relevant high-resolution 3D printing methods but is only rarely utilized in the multi-material 3D printing of materials. This study demonstrated multi-material stereolithography using combinations of materials with different Young’s moduli, i.e., 0.5 MPa and 1.1 GPa, for manufacturing pneumatic actuators and microactuators with a resolution as small as 200 μm. These multi-material actuators have advantages over single-material actuators in terms of their deformation controllability and ease of assembly.

## 1. Introduction

Actuators are one of the most vital components for a broad range of applications, such as soft robots [1], microfluidics [2], or biomedical engineering [3]. Recent efforts in the research of flexible actuators focused on actuation materials, such as shape memory polymers [4] or alloys [5,6], stimuli-responsive materials [7,8], electroactive polymers [9,10], and pneumatic-driven actuators [1,11,12]. Among these, pneumatic actuators based on the inflation of extensible structures attracted great attention due to their practicality [1], safety, and human friendliness, combined with the clear application potential. Therefore, the concept of pneumatic actuators became one of the frontier hot spots for commercial soft robots, e.g., the so-called BionicSoftArm from Festo AG & Co. KG [13].

Flexible pneumatic actuators are mostly made of polymer materials, such as silicone rubber [14] and thermoplastic polymers [15]. Although traditional manufacturing methodologies, such as molding [16], are the most frequently utilized, these techniques are inherently difficult to apply due to various process-inherent challenges, including differences in shrinkage from thermal retraction, adhesion with inclusion material, and difficulty in fabricating internal void volumes, e.g., the inflatable structures. The latter has been particularly challenging to overcome [12]. As a consequence, almost every pneumatic actuator is currently manufactured as a component that consists of several sub-components that have to be integrated after manufacturing. This makes the production process challenging and time-consuming. In recent years, there has been a focused interest in additive manufacturing techniques, also known as 3D printing, as an advantageous technique for rapidly fabricating components with high complexity [17,18]. Most 3D printing methods, including fused deposition modeling (FDM) [17,19], Polyjet [20,21], direct ink writing (DIW) [22,23], and SL [24,25], were found to be suitable for producing elastomeric pneumatic actuators. In 2019, Skylar-Scott et al. [22] reported the fabrication of multi-material actuators using DIW printing with multiple nozzles for printing pneumatic actuators from soft and rigid components.

However, DIW and FDM printing are extrusion-based methods, which involve building 3D structures from several strands that are extruded through a nozzle. These strands can have a minimum resolution of around 100 µm. Nevertheless, usually, only mesh-like structures with high resolution are built from these strands since real 3D structures cannot be built with high resolution. Polyjetting [26,27,28,29] is also used in multimaterial printing but cannot be used in the printing of features smaller than 0.75 mm (see Appendix A).

On the other hand, stereolithography offers printing resolutions down to the micron scale [30] but has so far been difficult to apply in multi-material printing. However, using stereolithography to print hard and soft materials would allow for the 3D printing of pneumatic actuators at high resolution and fidelity. Although stereolithography can be used to print flexible materials, such as aliphatic urethane diacrylate (AUD) [18,25], poly(trimethylene carbonate) [31], and hydrogels [32], with Young’s moduli as low as 0.58 MPa and a stretchability of 1100% [25], pneumatic actuators require not just soft materials but the combination of soft and hard materials. The latter is required for reinforcing stress-concentrated parts and interfaces of the actuators to the outside world. So far, this combination has not been demonstrated in stereolithography-printed multi-material actuators.

In this study, we present the combination of a hard and a flexible material resin formulation that allows for two-material printing via high-resolution stereolithography. The material formulation for the flexible elastomer was based on poly(ethylene glycol) dimethacrylate (PEGDMA) and acrylic acid (AA), while the resin for the rigid part consisted of diacylated and tetraacylated crosslinkers (Figure 1A). We demonstrated that multi-material printing of these two resins was possible by changing the vat during the printing process. The process flow for this multi-material printing is shown in Figure 1B. In the first step, the part was printed using one of the resins. In the second step, the resin bath was removed and the part was soaked and washed in a solvent. In the third step, the remaining component was printed using the second resin. Since the resin formulations for both the hard and soft parts contained a high concentration of acrylate monomers, both formulations could be covalently bonded to each other, ensuring a tight fit. Therefore, strong interlinking was achieved between the two materials, thus allowing for repeated stresses of the elastic component without risking delamination. We showed that this technique is capable of fabricating multi-material actuators and microactuators with a minimum feature size as small as 200 µm. The printed multi-material innovative structures, such as self-sealing balloons and pneumatically switchable surfaces, have the potential to be used as soft actuators or robots.

## 2. Materials and Methods

### 2.1. Materials

Poly(ethylene glycol) (PEG) (Mn~10,000), PEG (Mn~400), Pentaerythritol ethoxylate (15/4 EO/OH) (Mn~797), methacrylic anhydride, acrylic acid, 4-dimethylaminopyridine (DMAP), PEGDA (Mn~575), isobornyl acrylate, diurethane dimethacrylate (DUDMA), diphenyl(2,4,6-trimethylbenzoyl) phosphine oxide (TPO), dibutyltin dilaurate, and pentaerythritol tetraacrylate (PETA) were purchased from Sigma-Aldrich (Taufkirchen, Germany). Tetrahydrofuran, diethyl ether, and 2-propanol were purchased from Carl Roth. Tinuvin-326 was kindly provided by BASF (Ludwigshafen, Germany).

### 2.2. Synthesis of PEGDMA (Mn~10,000)

Unmodified PEG (Mn~10,000), methacrylic anhydride, and 4-dimethylaminopyridine (DMAP) were utilized to synthesize the linear crosslinker PEGDMA. The chemical synthesis of PEGDMA was performed as follows: In a 1000 mL round bottom flask, 100 g of PEG (10 mmol, 1 eq) and 300 mL of anhydrous tetrahydrofuran were stirred, and the flask was heated with an oil bath with a temperature of 50 °C. After the PEG was dissolved, 3.08 g of methacrylic anhydride (20 mmol, 2 eq.) and 61 mg DMAP (0.5 mmol, 0.05 eq.) were added, and the mixture was heated to reflux for 16 h. After the reaction, the solvent was removed under vacuum, and the rest of the mixture was precipitated in 1.5 L cold diethyl ether three times to remove all residual methacrylic acid and DMAP. The white solid was collected and dried in a vacuum oven at 30 °C and 0.1 mbar for 20 h; 96.3 g PEGDMA was thus received (yield: 94.8%). The synthesized PEGDMA was characterized using a Fourier transform infrared (FTIR) spectrometer (Frontier 100 MIR-FTIR, Perkin Elmer, Germany). The peak at 1670–1700 cm^−1^ indicated the expected carbon–carbon double bonds in the PEGDMA (see Appendix A).

### 2.3. Preparation of the Soft and Hard Resins

To produce the soft and hard resins, the chemicals listed in Table 1 and Table 2 were added to corresponding glass containers. The mixtures were placed in an ultrasonic bath for 30 min, after which clear resins were received. Two versions of the soft resin were prepared: a clear colorless version for transparent actuator printing and an orange-colored version used for tensile test measurements, as it allowed for significantly better observation of the two-material interface using the boundary line of the two colors.

### 2.4. Mechanical Performance Characterization

Compression and strain–stress tests were performed using an Inspekt Table 5 universal tensile testing machine (Hegewald & Peschke Meß- und Prüftechnik GmbH, Nossen, Germany) with a 100 N load cell. We characterized the yield strengths and Young’s moduli of 5 printed samples using a strain rate of 0.5 mm s^−1^ on a dogbone-shaped sample with a size of 10 mm × 4 mm × 1 mm. For the cyclic loading test on the dogbone-shaped samples, a maximum strain of 300% with a strain rate of 1 mm s^−1^ at an interval of 10 s during cyclic loading was applied. For the compression test, the components were first compressed to 80% of the thickness of the soft part and then released at a rate of 0.5 mm s^−1^. For each mechanical performance characterization, five samples were tested to calculate yield strengths and Young’s moduli.

### 2.5. Stereolithography

All stereolithography printing was done using the vat-based stereolithography printer Asiga Pico 2 (Asiga, Erfurt, Germany) with a lateral resolution of 50 µm. A dosage calibration step was required to establish the exposure time, which is one of the most critical printing parameters. A layer of the freshly prepared resin was first spread onto a glass slide and spots with a diameter of 3 mm were exposed to light with the same energy density as printing for different time steps from 4 to 15 s. After the uncured resin was removed, the glass slides were dried in nitrogen, after which the thickness of each dot was measured. Since the applied energy density was set to 8.87 mW cm^−2^, the corresponding exposure energy was calculated as follows:(1)$$E_{exposure}=I_{light}\;\cdot \;t_{exposure}\;$$

The layer thickness was plotted against the exposure energy, which allowed for determining the required exposure time for a given printing layer thickness. The soft material was printed with an energy density of 8.87 mW cm^−2^, a layer thickness of 50 µm, a z-compensation of 0, and a separation distance of 16 mm to make sure the printed structure could separate from the built tray bottom. An exposure time of 3.8 s was used for each layer. The hard material was printed with an energy density of 8.87 mW cm^−2^, a layer thickness of 50 µm, a z-compensation of 0, a separation distance of 6 mm (if the hard material was printed alone) or 16 mm (if hard material was printed on top of an existing layer of soft material [28]), and an exposure time of 4.3 s.

In multi-material printing, the printing parameters of each material were not changed. It is worth noting that after one of the materials was printed, the printed part was soaked in a solvent, e.g., ethanol or 2-propanol, for 20 s and then dried over nitrogen to remove residual resin to continue printing the second material. For more complex structures, the solvent was stirred to facilitate the development of non-cured resin. Since the base platform was not moved after the first material was printed, no alignment problem needed to be compensated for. After the resin vats were changed, the second material was printed in the same condition as it was printed alone, where no special overlap, burn-in, or other steps were required.

After printing, the parts were flushed with isopropanol and ethanol. Internal surfaces of the balloons or actuators were washed by injecting solvent into them with a syringe. After washing, the parts were placed in an XYZprinting UV Curing Chamber (UV LED, 375–405 nm) and post-cured with an exposure energy density of 0.2 W cm^−2^ for 10 min. Afterward, the components were washed with water, isopropanol, and ethanol in an ultrasonic bath for 10 min. Finally, the components were dried in a vacuum oven at 100 °C and 0.1 mbar for one hour.

## 3. Results and Discussion

### 3.1. Mechanical Assessment of the Multi-Material Components

The soft resin had a low viscosity at room temperature (308 mPa·s at 25 °C) and was thus appropriate for stereolithography. We first printed dogbone-shaped components for the characterization of Young’s modulus and the elongation at yield using stress/strain measurements. We were able to stretch the material up to 589% ± 59%, finding an initial Young’s modulus of around 0.52 ± 0.06 MPa according to five parallel tensile tests, indicating that the softness and stretchability were suitable for flexible pneumatic actuators (Figure 2A) [33]. To be sure that the elastic properties of the resin were stable over time and not due to incomplete polymerization, the resin was left under UV exposure for different periods. Using an XYZprinting UV Curing Chamber with an exposure energy density of 0.2 W cm^−2^ at 375–405 nm we post-cured the dogbone-shaped components for 10, 20, 30, and 40 min. As can be seen from Figure 2B, the maximum exposure (i.e., 40 min) caused only a slight change in Young’s modulus (Figure 2B, line) and the yield strain (Figure 2B, bars). It is expected that the actuators made of this material are stable under exposure to sunlight.

The hard resin consisted of higher concentrations of multi-acrylate terminated crosslinkers, including DUDMA and PETA. The prepared resin showed a viscosity of 70 mPa·s at 25 °C. The stress/strain curve is shown in Figure 2C, from which Young’s modulus of 1.1 GPa was derived, as well as a yield strain of 6%.

To demonstrate the feasibility of multi-material 3D printing, we printed multi-material dogbone-shaped components and Kelvin foam lattice structures. The structures were assessed via stress/strain and compression testing. As can be seen in Figure 2D, no delamination occurred at the interface between the soft and hard material, even after stretching by up to 500%. Furthermore, we found that bulk failure within the soft segments at yield strains around 550% was the primary failure mode (Figure 2E). Cyclic load testing was performed for the multi-material dogbone-shaped components, whereby the component was stretched up to 300% yield for 300 cycles without interface damage (Figure 2F). Finally, in the compression test of the printed Kelvin foam lattice, no interface cracks showed up within the compression distance of 1.5 cm, which was equal to 75% of the soft part, and the rebound was synchronized with the back-lifting (Figure 2G, Appendix A).

### 3.2. Three-Dimensional Printing of Multi-Material Actuators

We tested a variety of multi-material pneumatic actuatable structures to demonstrate the feasibility of our multi-material manufacturing approach for these applications. We first designed a balloon with a rigid air inlet. This prototype helped to confirm that the rigid material could fix the air inlet, i.e., the air inlet would not enlarge or flip down as the balloon body expanded. Figure 3A illustrates the design of the balloon with a diameter of 20 mm and a wall thickness of 2 mm. The rigid component was a 0.5 mm thick annulus with inner and outer diameters of 1 mm and 3 mm, respectively. After the balloon was printed and washed, we inflated the balloon with nitrogen gas and expanded the balloon to a diameter of 80 mm (Figure 3B). As shown in Figure 3C, the rigid entry hole maintained its original shape while the balloon was inflated such that we could use tape to seal it (also see Appendix A). The tape-sealed balloon was nicely airtight such that the elastic nature of the balloon allowed it to jump back after being hit (see Appendix A). We further demonstrated that the system could be self-sealing by sealing the hard segment with another flexible material segment. By printing flexible material onto the surface of the rigid ring to seal the air inlet, the soft material itself was only stressed by the internal air pressure but not by the expansion of the balloon. This structure was achieved by adding a 1 mm-thick, 3 mm-diameter soft disk on the hard annulus of the multi-material balloon, as shown in Figure 3D. We printed this multi-material part and pressurized the balloon with the help of a 200-micron-diameter needle. After the needle was removed, the balloon kept the inflated shape, and the pinhole was also inconspicuous (Figure 3E). In order to ensure that no leakage happened, the filled balloon was subjected the compression testing (Figure 3F). The balloon did not show a significant decrease in stress when squeezed, indicating that no noticeable gas leakage occurred (Figure 3G). Furthermore, considering that a threaded rigid inlet may also be useful, we next printed additional multi-material actuators. Figure 3H shows a printed set of multi-material actuators with rigid threaded air inlets. It was thus possible to directly print the connectors, e.g., for an external gas reservoir supply with the actuator. Compared with traditional pneumatic actuators, this allowed for a significantly simplified assembly (see Appendix A). In our example, the designed set was assembled into a simple pneumatic gripper that was directly connected to a pressurized gas supply via a threaded connector (Figure 3I).

### 3.3. Three-Dimensional Printing of Multi-Material Microactuators

Miniature pneumatic systems are widely employed in microvalves, micropumps, and other control systems [34]. We also printed two multi-material microactuators to demonstrate the potential applications of multi-material components for small-scale systems. The first type was a rigid chip with a soft membrane (Figure 4A). An array of holes was printed with the hard material and a flexible membrane was printed on top so that only the soft part on the top can be inflated (Figure 4B). The air inlet was manufactured on the side of the chip from which gas pressure could be applied to the system to pressurize the cavity in the chip and actuate the soft membrane. After the film expanded, the surface changed into an uneven surface, and the surface morphology could be controlled pneumatically (Figure 4C). Based on this model, we placed rigid switchable fin-like structures on top of the membranes, as shown in Figure 4D. Here, the fin structures were designed to flip via pressurizing and deforming the membrane (Figure 4E). The results of the printing and actuation are displayed in Figure 4F, demonstrating the movement of the fins (see Appendix A).

With the herein proposed material system, it was thus possible to directly 3D print pneumatically responsive surfaces using stereolithography. Compared with other manufacturing methodologies, such as soft lithography, this approach allows for combining hard and soft materials for a significantly enhanced focus on the expansion motion, as well as a significantly simplified manufacturing process.

## 4. Conclusions

In this study, we demonstrated the multi-material 3D printing of pneumatic actuators using high-resolution SL. We developed a novel two-resin printing formulation with a soft resin made from a PEG-AA-based elastomer for the flexible sections of the actuator. This material exhibited excellent stretchability with yield strains around 550% and high stability in UV light. The hard resin consisted of crosslinked acrylates and exhibited supporting and fixing parts. The two resins could be printed in the stereolithography instrument, forming a strong interface bonding whose yield strength exceeded the strength of the soft material. We demonstrated the feasibility of this approach for the generation of multi-material soft actuators using a balloon with a self-sealing rigid air inlet. We showed that threaded connectors to pressurized gas lines could be directly 3D printed, which significantly simplified the overall system setup. We demonstrated that this approach could be used to directly 3D print common types of actuators used in microsystem technology, including soft membrane actuators and actuators for surface motion, as demonstrated using a fin-type actuatable surface.

## Figures and Tables

**Figure 1 micromachines-14-00244-f001:**
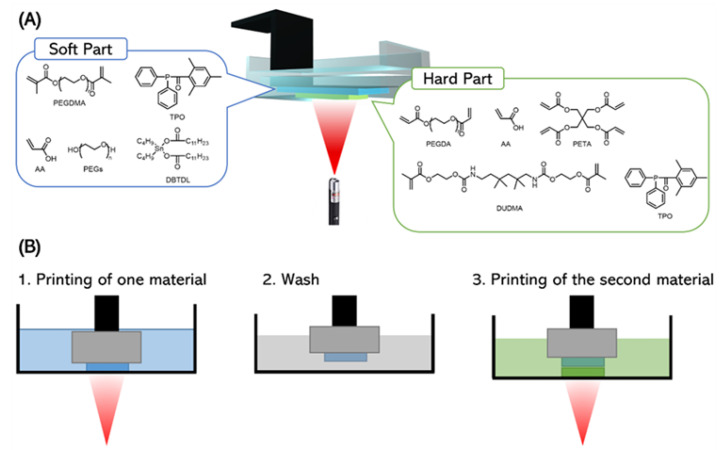
Concept of the multi-material printing system. (**A**) The resin composition for the soft (blue) and rigid material (green). (**B**) Schematic illustrations of the process flow for multi-material printing using three steps: printing with the first material, washing with solvent to remove the residue resin, and printing with the next material.

**Figure 2 micromachines-14-00244-f002:**
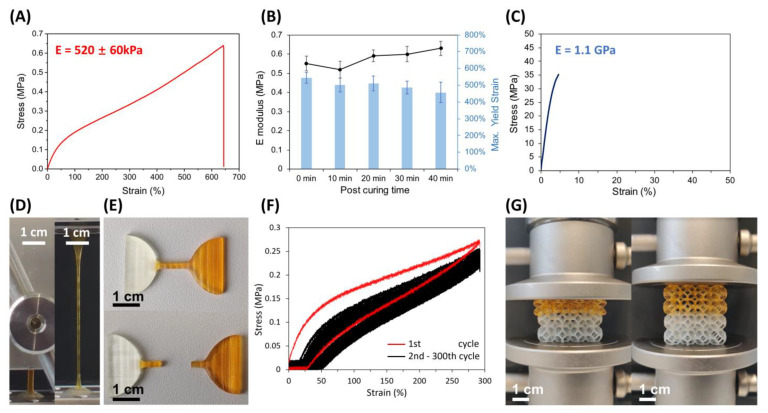
Mechanical characterization of the soft and multi-material components. (**A)** The stress/strain curve of the printed soft material with a calculated Young’s modulus of 520 ± 60 kPa. (**B**) The maximum yield strain (bars) and Young’s modulus (line) of printed parts after different post-curing periods. (**C**) The stress/strain curve of the printed hard material with the calculated Young’s modulus of 1.1 GPa. (**D**) The multi-material component before and after 500% straining, with a stretched part size of 1.5 mm × 1.5 mm × 8 mm. The orange section (top) is the PEG-AA-based soft material, while the colorless section (bottom) is the hard material. (**E**) An overstretched multi-material component demonstrated that the component broke within the bulk and not at the interface. (**F**) The stress/strain curve of 300 rounds of cyclic loading of the multi-material component in (**D**) showed no cleavage or significant change in Young’s modulus during the repeated stretching. Here, the red line refers to the first cycle and the black line refers to the rest of the cycles. (**G**) Multi-material Kelvin’s foam (side length 4 cm). The left image shows the morphology of the lattice when compressed by 1.5 cm (75% of the soft part), while the right image shows the instantaneous rebound upon release (as in Appendix A).

**Figure 3 micromachines-14-00244-f003:**
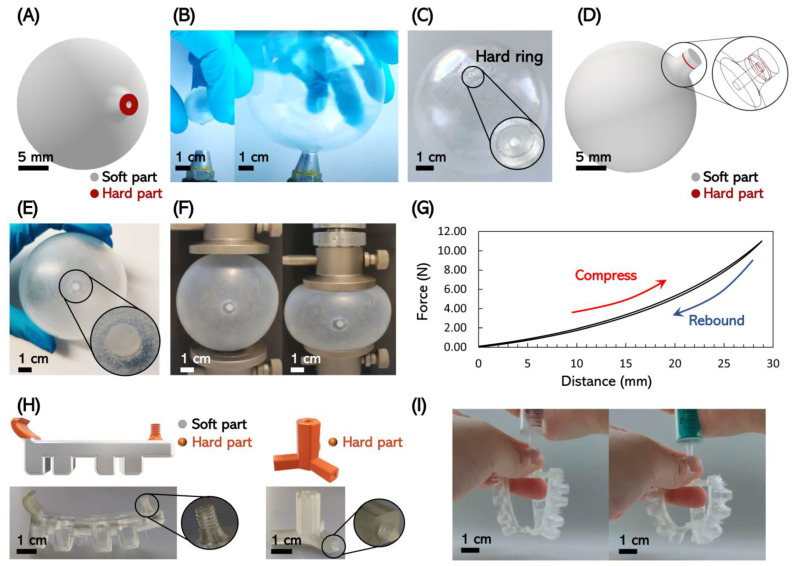
Three-dimensional printing of multi-material actuators. (**A**) Design of a multi-material balloon (diameter = 20 mm; thickness = 2 mm; diameter of the inlet: outside = 3 mm, inside = 1 mm; thickness of the hard disk (red part) = 0.5 mm). (**B**) Printed balloon in the non-inflated and inflated states. (**C**) Inflated balloon with a hard ring to fix the inlet, which was sealed using tape, as the hard ring was not deformed. (**D**) The design of a multi-material balloon with a soft valve as a lumen subjected to compression testing. (**E**) Balloon with a soft valve that had already been inflated with a needle, and no obvious pinhole existed in the soft valve. (**F**) Compression tests of an inflated balloon with this valve demonstrated that no air leaked from the balloon. (**G**) The curve of the compression test of the (**F**) process, where the agreement of the compression and rebound curves illustrated the airtightness. (**H**) Multi-material actuator with a hard, 3D printed threaded inlet for a pressure supply connector directly manufactured into a soft gripper and the hub structure with threaded sleeves, which could be assembled with actuators. (**I**) The assembled pneumatic gripper in non-actuated (left) and actuated states (right).

**Figure 4 micromachines-14-00244-f004:**
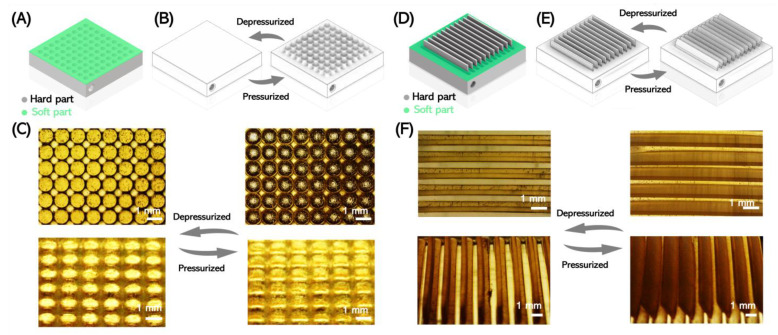
Three-dimensional printing of microactuators. (**A**) Design of a microactuator chip with a threaded pressurized air inlet on the side. The lateral size of the chip was 20 mm × 20 mm × 3 mm, with a soft membrane that was 200 µm in thickness. (**B**) The unactuated state had an even surface; after actuation, the membrane expanded and the inflated sections became visible. (**C**) The printed structure (left) and the actuated state after pressurizing (right). (**D**) Design of a microactuator fin-chip. The lateral size of the chip was 20 mm × 20 mm × 3 mm, with the fins being 14 mm in length, 0.3 mm in thickness, and 2 mm in height. (**E**) The fins were initially in standing mode when the system was not pressurized and tipped over after pressurization. **(F**) The printed pneumatic-driven fins (left) and the actuated state (right).

**Table 1 micromachines-14-00244-t001:** Resin formulations for the soft material.

Component of Soft Resin	Orange Version Unit: wt.%	Colorless Version Unit: wt.%
PEGDMA (Mn~10,000)	10	10
PEG (Mn~4000)	20	20
PEG (Mn~400)	20	20
Pentaerythritol ethoxylate (Mn~797)	20	20
Acrylic acid	29.3	29.1
DBTBL	0.5	0.5
TPO	0.2	0.2
Sudan Orange G	0.02	-
Tinuvin-326	-	0.2

**Table 2 micromachines-14-00244-t002:** Resin formulations for the hard material.

Component of Hard Resin	Unit: wt.%
DUDMA	25
Acrylic acid	25
PEGDA (Mn~575)	25
PETA	24.6
TPO	0.2
Tinuvin-326	0.2

## Data Availability

The data used to support the study are available upon request to the corresponding author.

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
