# Peer review of "Fabrication of Multi-Material Pneumatic Actuators and Microactuators Using Stereolithography"

_micromachines, 2023, doi:10.3390/mi14020244_

Round 1
Reviewer 1 Report
This manuscript reports about the fabrication of pneumatic actuators by stereolithography using hard and soft materials. The presented approach is sound and of interest for a broad readership.
Author Response
Dear Editors and Reviewers,
we thank the reviewers for their expert input and detailed review of our work and the helpful suggestions they have made. We have addressed all the issues raised – please find our response to the comments of the reviewers on the following pages. The reviewer’s comments are set in italic whereas our comments are set in upright font. Changes to the manuscript are marked in yellow.
Remark 01:
This manuscript reports about the fabrication of pneumatic actuators by stereolithography using hard and soft materials. The presented approach is sound and of interest for a broad readership.
Answer: We thank the reviewer for this positive assessment of our work.
Reviewer 2 Report
This is a well structured and nicely written paper adressing the important issue of fabricating pneumatic actuators by multimaterial additive manufacturing.
There are only a few technical remarks and remarks with respect to language.
Technical Remarks
- Section 2.5: I assume that the washing step is critical in particular with respect to clearing internal volumes from unused resin. The soaking times presented in section 2.5 seem very short. Does this also apply to the balloon or to the microactuators in section 3.3 or is supporting actuation of the solvent e.g. by ultrasound required for closed small structures?
- Section 3.1: Please specify how many samples have been tested in each mechanical characterisation.
Remarks wrt. language
- I recommend to consistently use present tense in the entire paper
- There are a few typos left
- The headings of table 1 and 2 should be made unambiguous
- The term „domino-chip“ for the design in figure 4D is confusing. It looks rather like switchable fins.
Reviewer 3 Report
The manuscript from Song and coworkers reported a simple but useful method to create hybrid (hard and soft) 3D printed structures. The manuscript can give us informative insights on of new approaches to creating actuators and soft robots. I recommend this manuscript for publication in Micromachines, but the authors are recommended to revise the manuscript and address the following minor issues.
11) The titles of Table 1 and 2 should be revised to reflect the corresponding content.
22) Maybe replace SL with stereolithography since it is only one word to avoid acronym-heavy reading.
33) The soft part and hard part in Figure 1A is unclear.
44) More explanation can be added to the sentence “Due to …two materials” in line 84 and 85.
55) The quality of Figure 2 A, B and C can be improved.
